# SemJoin: Semantic Join Optimization

Christopher Gou
Purdue University
cgou@purdue.edu

Aditya Banerjee
Purdue University
banerj59@purdue.edu

Jiaxuan Wang
Purdue University
wang5433@purdue.edu

Chunwei Liu
Purdue University
chunwei@purdue.edu

## ABSTRACT

Integrating unstructured data into relational database systems is increasingly important as demand grows for natural language querying and analysis. A *semantic join*, joining two tables under a natural-language predicate, can be evaluated with a large language model (LLM), but comparing every pair of tuples requires $O(M \times N)$ LLM invocations and is cost-prohibitive at scale. Existing systems reduce this cost but typically commit to a single fixed strategy (e.g., embedding similarity or one batched scheme) regardless of the data or the join predicate. We propose an LLM-agent-based decision pipeline that optimizes semantic joins by matching the execution strategy to the characteristics of the underlying tables. An LLM advisor routes each join to one of two strategies: a *Cluster Join*, which prunes candidates via unsupervised embedding clustering and sample-based filtering, or a *Classifier* strategy for predicates that reduce to a shared discrete label set. Across three diverse datasets (IMDb reviews, email contradictions, and Stack Overflow tags), the advisor consistently identifies the optimal execution strategy for each workload. This dynamic routing proves decisive: it outperforms adaptive block join (ABJ) by 20–33 F1 points across all datasets while consuming fewer tokens on two of the three, and achieves higher F1 scores than featurized-decomposition join (FDJ) at one to two orders of magnitude lower token cost.

**VLDB Workshop Reference Format:**
Christopher Gou, Aditya Banerjee, Jiaxuan Wang, and Chunwei Liu. SemJoin: Semantic Join Optimization. VLDB 2026 Workshop: NOVAS.

**VLDB Workshop Artifact Availability:**
The source code, data, and/or other artifacts have been made available at https://github.com/Kronk12/CS-541-Semantic-Join-Exploration.

## 1 INTRODUCTION

Relational database systems are foundational to modern computing. Standard `JOIN` operations use predicates that rely on strict logical relationships, such as exact string matches or simple numerical comparisons [6]. While very effective for highly structured data, this architecture struggles with unstructured data, which typically requires semantic queries to be expressed in terms of natural language [2, 4, 5]. For example, a traditional database cannot execute a query to join a table of shirts with a table of pants based on the natural language predicate: "the items match as an outfit."

The advancement of Large Language Models (LLMs) in recent years presents a solution to this limitation. Using LLMs, we can

perform semantic joins, in which tables are joined using a predicate expressed in natural language [11]. A naive way to do so is to simply iterate over all pairs of items ("tuples"), where one item belongs to each table. However, this approach is inefficient and expensive. For two tables of size $M$ and $N$, this results in $O(M \times N)$ LLM invocations. At scale, this naive approach quickly becomes impractical due to the cost, both in terms of LLM token usage and time.

This paper explores optimization strategies that make semantic joins computationally feasible without sacrificing accuracy. Rather than committing to a single fixed algorithm, we introduce an LLM-driven *advisor* that inspects the join predicate and a small sample of each table and routes the join to the strategy best suited to the data: a *Cluster Join*, which uses unsupervised embedding-based clustering and sample-based filtering to prune the candidate space, or a *Classifier* strategy for predicates that reduce to a shared discrete label set. Every decision has a deterministic fallback, so the pipeline always reaches a valid configuration. We evaluate the resulting system on three datasets against two recent baselines. The first is Adaptive Block Join (ABJ) [9], a block-nested-loop scheme that packs batches of tuples from both tables into a single prompt and adaptively sizes those batches without knowing the predicate's selectivity in advance; it is the approach our Cluster Join is most directly designed against, and the strongest prior method we are aware of for the batched-prompting regime. The second is Featurized-Decomposition Join (FDJ) [10], a recent method that uses an LLM to extract features into a conjunctive-normal-form decomposition and prunes pairs with cheap distance functions, providing statistical guarantees on the precision and recall of its output.

In summary, this paper makes the following contributions: **(1)** We propose **SemJoin**, an LLM-agent pipeline that treats strategy selection as the core problem: an LLM advisor inspects the join predicate and a small sample of each table and routes the join to the execution strategy best matched to the data, with deterministic fallbacks that guarantee a valid configuration at every decision. **(2)** We instantiate the pipeline with two complementary execution backends—a *Cluster Join* that prunes candidate pairs via embedding clustering and sample-based cluster-pair filtering, and a *Classifier* for predicates that reduce to a shared discrete label set—and give the advisor concrete, sample-driven criteria for choosing between them, with the Cluster Join serving as the universal fallback. We further provide an optional projection step for cross-table embedding alignment. **(3)** We evaluate the pipeline on three datasets against ABJ and FDJ, a recent guarantee-based method. The routed strategy exceeds ABJ's F1 score on every dataset while reducing token cost on two of three, exceeds FDJ's F1 on all three at far lower token cost, and the advisor selects each workload's optimal strategy at negligible overhead.

## 2 RELATED WORK

In this section, we review prior work in two areas on semantic joins: *AI database systems*, which embed semantic operations within full engines, and standalone *join-optimization techniques*, which focus on semantic joins.

### 2.1 AI Database Systems

A growing number of AI database systems (AIDBs) extend the relational model with "semantic operators" parameterized by a natural-language expression and evaluated through the use of LLMs. Beyond a single operator, each of these is a *system*: it exposes a suite of semantic operators (e.g., filter, join, map, aggregate, top-k, group-by) through a declarative interface, and ships an optimizer that selects physical implementations. LOTUS, DocETL, Palimpzest, iPDB, and Cortex AISQL are all examples of such systems, each supporting semantic joins among several other operations [2–4, 7, 8]. DocETL uses an LLM agent as a query optimizer, rewriting LLM-powered operator pipelines for complex document processing [8]. Palimpzest casts AI workloads as relational views computed via a `convert` operator and uses a cost optimizer to trade off runtime, financial cost, and output quality [4].

LOTUS exposes its operators through a DataFrame-based API and pairs each with an optimization framework that provides statistical accuracy guarantees relative to a high-quality "gold" algorithm [7]. For semantic joins, its gold algorithm is a tuple nested-loop evaluation, which LOTUS optimizes by dynamically selecting between two cheaper proxy algorithms. The first, *sim-filter*, directly evaluates the embedding similarity between join keys. The second, *project-sim-filter*, invokes an LLM over each tuple in the left table to predict the expected value of the right join key, and then evaluates embedding similarity. By sampling a small fraction of the dataset, LOTUS calculates thresholds for each proxy, allowing it to send ambiguous matches to the expensive LLM while relying on inexpensive embedding similarity for obvious matches and rejections.

Snowflake's Cortex AISQL integrates LLM-based semantic operators into SQL and makes the core insight that many semantic joins are functionally equivalent to multi-label classification problems [3]. Instead of executing a naive semantic join, it takes a tuple from one table and classifies it against the values of the other table, treating those values as a set of candidate labels. By evaluating all potential matches for a tuple in a single prompt, the system reduces the number of required LLM calls from quadratic to linear complexity. An AI oracle is given the natural-language prompt, schema metadata, table sizes, and sample values from both tables to determine when such an application is possible. The approach is most useful for asymmetric tables where the 'label' table is small, because a large label set must be batched across prompts, reintroducing quadratic complexity.

### 2.2 Optimizing Semantic Joins

In contrast to the systems above, two recent works are standalone techniques for optimizing the semantic join operator itself, rather than full query engines. These are most directly related to our approach.

*Adaptive Block Join.* ABJ explores the optimization strategy of packing batches of tuples into each LLM prompt to take advantage of the context window in modern LLMs [9]. ABJ packs batches of tuples from both tables into a single prompt, a block nested-loop evaluation, and instructs the LLM to output all matching pairs in the batch. The effective batch size depends on the predicate's selectivity, as a more selective predicate produces more output tokens, meaning that fewer tuples can be packed into the prompt before it overflows the token limit. ABJ begins with an optimistic (low) selectivity estimate and iteratively lowers the batch size when an overflow occurs, attaining near-optimal batch sizes without knowing the selectivity of the join predicate in advance.

*Featurized-Decomposition Join.* FDJ observes that existing model-cascade solutions, which rely purely on embedding-based semantic similarity, often yield limited gains in practice [10]. This is because text records are often lengthy and contain both relevant and irrelevant information, causing standard embeddings to poorly encode the specific, logical join criteria. To address this, FDJ uses LLMs to automatically extract reliable features from the unstructured text and combines them into a logical expression in conjunctive normal form (CNF). This "featurized decomposition" evaluates the join condition using inexpensive distance functions (lexical, arithmetic, or embedding-based) on the extracted features, drastically reducing the need for LLM calls on every tuple pair. FDJ further provides tight statistical guarantees on the precision and recall of the output by carefully setting threshold parameters for these predicates using labeled samples.

Our Cluster Join methodology shares FDJ's underlying philosophy of intelligently pruning the search space to minimize expensive pairwise LLM comparisons. However, while FDJ achieves this through explicit feature extraction and logical predicate formulation, our approach relies on unsupervised embedding-based K-Means clustering.

## 3 METHODOLOGY

In this section, we present our approach to optimizing the semantic join operator. Our system is structured as an LLM-agent-based decision pipeline: rather than following a single fixed algorithm, our system uses an LLM-driven advisor to analyze the join predicate and the data tables and dynamically make decisions about the strategy for performing the join and the embedding model(s) used. At runtime, the advisor chooses one of two strategies to perform the join: a *classifier* strategy for predicates that reduce to equi-joins on a discrete set of labels, and a *cluster join* strategy for all other semantic predicates. Every decision has a deterministic fallback (e.g., ambiguous strategy defaults to cluster join, unrecognized embedding model defaults to `all-mpnet-base-v2`), ensuring that the pipeline always reaches a valid configuration. Additionally, optional parameters allow users to override any advisor decision. An overview of our approach can be seen in Figure 1.

### 3.1 Strategy Selection

In every semantic join invocation, the first step is strategy selection. The join predicate, the schemas of both tables, and a small sample of tuples from both tables are sent to the LLM, which is then asked to choose one of the two semantic join strategies:

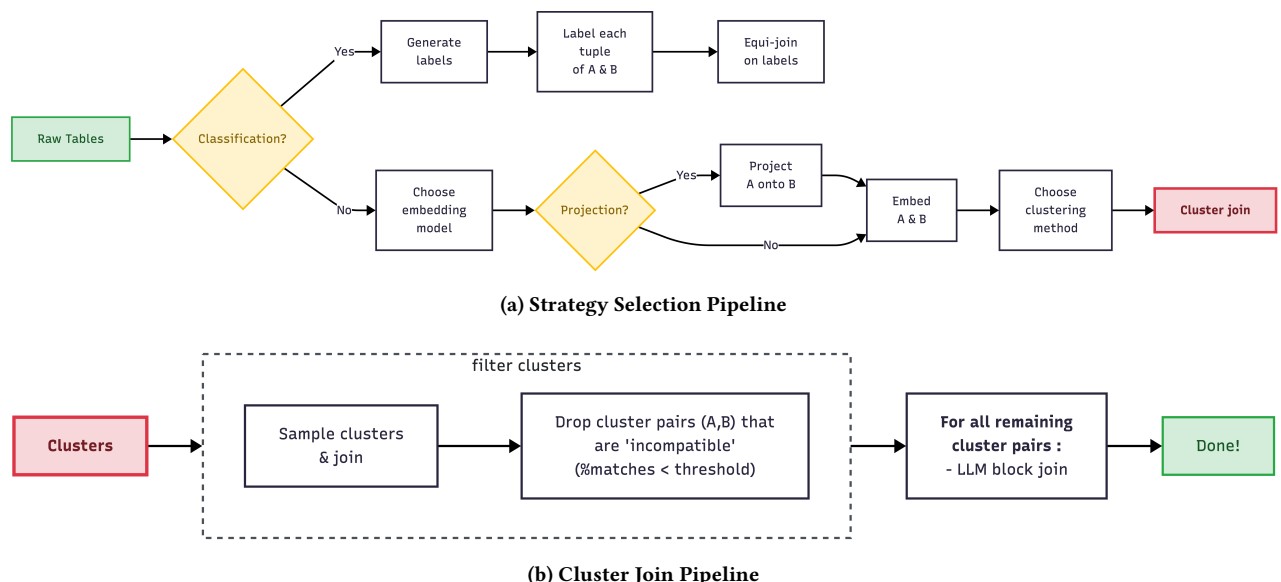

**(a) Strategy Selection Pipeline**

**(b) Cluster Join Pipeline**

**Figure 1: Overview of our approach: (a) the strategy selection pipeline and (b) the cluster join pipeline.**

(1) **Classifier**: The predicate is a "same-label" join over a small set of discrete classes/labels (e.g., same sentiment, same genre). In this case, each tuple can be independently assigned a label, and matching reduces to an equi-join on these labels.

(2) **Cluster Join**: The predicate is not well-suited to the above classifier strategy and requires directly comparing each pair of tuples.

If the LLM advisor returns an ambiguous result, our system defaults to the cluster join strategy, which serves as the universal fallback for any predicate that the advisor cannot confidently categorize. However, users can bypass the advisor and specify the strategy via the force_strategy parameter.

## 3.2 Classifier Strategy

For predicates that reduce to "same-label" joins, the semantic join is performed using the classifier strategy as follows: First, the LLM is given a sample of Table *B* and generates the list of canonical labels (e.g., ["positive", "negative", "neutral"]). Next, both tables are partitioned into batches and sent to the LLM, which assigns exactly one of the canonical labels to each tuple. Finally, a simple equi-join is performed between *A* and *B* on these labels.

This strategy can be seen as a generalization of AISQL's classification optimization as if the semantic join can be converted into a multi-label classification problem then the join is treated as a "same-label" join where the 'label' table can be used to generate the list of canonical labels.

Note that an "unknown" label is always included, and is assigned to tuples that the LLM cannot confidently classify. During the equi-join, tuples labeled "unknown" in either table are excluded from matching.

## 3.3 Cluster Join Strategy

For predicates that require direct comparisons of tuple pairs, the cluster join strategy executes the following stages: embedding model selection, optional projection, embedding and clustering, cluster-pair filtering, and LLM-based matching.

*3.3.1 Embedding Model Selection.* First, the LLM advisor selects the most appropriate sentence-transformer model from the given options (e.g., all-mpnet-base-v2, all-MiniLM-L6-v2). To make this decision, the LLM is again given the join predicate, both table schemas, and a sample of tuples from both tables.

*3.3.2 Optional Projection.* Before generating embeddings, the LLM advisor decides whether a *projection step* is needed. Projection is warranted when independent clusterings of Tables *A* and *B* may not correspond well to each other, e.g., when semantic similarity between tuples in *A* does not indicate that they will be matched to semantically similar tuples in *B* under the join predicate. This step was added due to the concern of using semantic similarity as part of our filter step in the Cluster Join pipeline and is inspired by LOTUS' *project-sim-filter*.

When projection is enabled, tuples of Table *A* are batched and sent to the LLM along with a sample of Table *B* and the join predicate. The LLM is asked to predict what the matching Table *B* tuple would look like for each Table *A* tuple, producing projected text strings. These projections are then used in lieu of the original Table *A* for the embedding step.

As a concrete example, consider the Stack Overflow dataset described in Section 4.2.3. Table *A* contains verbose questions (e.g., "I am trying to figure out oAuth2 ... ValueError at /api/customer/...") while Table *B* contains single-word concept labels (e.g., "javascript"). Without projection, the two tables cluster independently over very different kinds of text, producing clusterings with potentially very little correspondence. With

projection, Table $A$'s tuples are first transformed into single-word concept label predictions. As a result, both tables are clustered over the same kind of text, allowing for much closer correspondence between clusters across the two tables. This allows the later filter stage to find and filter out unrelated cluster pairs before the final matching stage.

*3.3.3 Embedding and Clustering.* Each tuple of both tables is serialized into a single text string by concatenating its attributes and separating them by a delimiter character (|), with each field truncated to a maximum number of characters (400 by default). The selected sentence-transformer model generates embeddings for each of these strings.

Next, the two tables' embeddings are clustered independently, with the advisor choosing the appropriate clustering algorithm from the given options (e.g., $k$-means, HDBSCAN). After clustering both tables, we enumerate all pairs $(c_A, c_B)$ of an $A$ cluster and a $B$ cluster. Each cluster pair represents a block of tuples from each table that will be evaluated together in the final LLM join step.

*3.3.4 Cluster Pair Filtering.* The next step then filters out incompatible cluster pairs before the expensive LLM join. For each cluster pair, a random sample of tuples is drawn from each table and a small LLM join is performed on this sample, producing a set of matches. The match rate is computed as follows:

$$\text{match rate} = \frac{\text{number of matches}}{(\text{size of sample } A) \times (\text{size of sample } B)}$$

If the match rate is lower than the configured `filter_threshold`, the cluster pair is discarded. Note that if either cluster has fewer tuples than the configured `min_profile_size`, the cluster pair is immediately retained and does not go through the filtering stage, since the cost of the small LLM join would approach the cost of simply performing the join between the entire clusters.

*3.3.5 LLM-Based Matching.* For each surviving cluster pair, the LLM is used to perform the actual semantic join. The LLM is instructed to compare every tuple in Table $A$ against every tuple in Table $B$ and output all of the matches. Similarly to ABJ [9], we partition both clusters into batches and include an entire batch of each cluster's tuples in each LLM prompt in order to leverage more of the LLM's available context window.

## 4  EVALUATION

In this section, we evaluate **SemJoin** on three datasets against the ABJ and FDJ baselines. We additionally present parameter studies on individual components.

### 4.1  Baseline

Our primary baseline is ABJ [9], one of the strongest prior approaches we are aware of. Because our Cluster Join is designed as a cost-efficient alternative to this method, replicating its behavior was one of our goals. Additionally, we also benchmark against FDJ [10], a recent state-of-the-art approach to semantic join optimization.

### 4.2  Datasets

*4.2.1 IMDb Movie Reviews.* For our initial dataset, we decided to mimic the "Reviews" experiment from ABJ's paper [9]. To paraphrase from the paper, "The second scenario ('Reviews') is based on the IMDB movie reviews, available for instance on Kaggle [1]. The goal is to match reviews with similar underlying sentiment (the data set comes with ground truth labels, labeling reviews as either positive or negative)."

*4.2.2 Email Contradictions.* For our next dataset, we wanted to evaluate our pipeline across a broader variety of datasets and specifically test its performance on a scenario with a more complicated, logical join predicate. To achieve this, we adapted the "Emails" scenario introduced by ABJ [9]. Loosely based on the Enron scandal investigation, this dataset requires the LLM to identify inconsistencies between executive statements and employee emails using the natural language condition "the two texts contradict each other." The statements are structured as "[Name]: I first heard about the losses in February 2022," while the emails are formatted as "I first told [Name] about the losses [TimeFrame]". Our goal was to use this dataset to evaluate our cluster join methodology on a join predicate that required complex deductive reasoning rather than simple semantic similarity.

*4.2.3 Stack Overflow Questions and Tags.* For our final dataset, we wanted to test our Cluster Join pipeline on a semantic join where semantic similarity is a poor heuristic for filtering out tuples. This is because our filtering step was inspired by LOTUS' methodology which struggles when semantic similarity is not a reliable indicator to filter out tuples [7]. Thus, we decided to create a dataset where one table had large amounts of text and another table had little to no text. This inspired the creation of the Stack Overflow dataset which contains a table of Stack Overflow questions and another table of tags (e.g., java, html, c++, etc).

Table 1 summarizes the key statistics for each dataset. Selectivity is defined as the fraction of all $|A| \times |B|$ candidate pairs that satisfy the join predicate.

**Table 1: Dataset Statistics**

| Dataset | $|A|$ | $|B|$ | True Matches | Selectivity |
|---------|-------|-------|--------------|-------------|
| IMDb    | 50    | 50    | 1,220        | 48.80%      |
| Emails  | 100   | 100   | 577          | 5.77%       |
| Stack   | 250   | 10    | 275          | 11.00%      |

### 4.3  Results

Table 2 reports the recall, precision, F1, and token cost for each strategy on all three datasets, average over three trials, alongside the ABJ and FDJ baselines. A notable result was that the advisor-chosen strategy exceeded ABJ's F1-score on every dataset while reducing token usage on two of the three. The Cluster Join figures reflect the best per-dataset configuration from our parameter search (Section 4.4.1) and thus represent a tuned upper bound; the Classifier and advisor results involve no such per-dataset tuning.

**Table 2: Performance and token expenditure across join strategies, average over three trials. The** shaded **row is the LLM advisor's choice; ABJ and FDJ are the baselines. Best value per column per dataset is bold. Cluster Join rows ([‡]) report the per-dataset configuration selected by the grid search of Section 4.4.1 and should be read as a tuned upper bound; the Classifier and advisor results involve no such per-dataset tuning. [†]On Emails, contradiction is not a same-label predicate; the only available taxonomy (person name) yields a vacuous name-identity equi-join, so the Classifier's quality scores are artifacts and are omitted.**

| Dataset | Strategy | Recall (%) | Precision (%) | F1 (%) | Tokens |
|---|---|---|---|---|---|
| **IMDb** | Cluster Join[‡] | 41.39 | 84.45 | 55.56 | 38,624 |
| | **Classifier** | **75.84** | 80.46 | **78.08** | **22,362** |
| | ABJ | 39.35 | 69.69 | 49.92 | 70,446 |
| | FDJ | 26.87 | **98.40** | 42.20 | 5,296,336 |
| **Emails** | **Cluster Join[‡]** | 73.48 | 76.81 | **75.11** | 26,963 |
| | Classifier[†] | — | — | — | **7,432** |
| | ABJ | 32.29 | 60.10 | 42.00 | 43,340 |
| | FDJ | **100.00** | 10.03 | 18.27 | 1,554,914 |
| **Stack** | Cluster Join[‡] | 63.27 | 77.68 | 69.74 | 76,942 |
| | **Classifier** | 66.91 | **85.19** | **74.95** | 30,216 |
| | ABJ | 46.18 | 67.55 | 54.86 | **26,895** |
| | FDJ | 30.67 | 70.87 | 42.77 | 3,607,262 |

*Cluster Join vs. baseline.* Against ABJ, the Cluster Join improves F1 on all three datasets while also reducing token expenditure on Emails and IMDb. On Emails it reaches an F1 of 75.11% versus ABJ's 42.00% at 26,963 tokens versus 43,340, and on IMDb 55.56% versus 49.92% at 38,624 tokens versus 70,446. The one exception to this efficiency trend is Stack Overflow, where the Cluster Join used more tokens than ABJ—76,942 versus 26,895—while achieving a higher F1 of 69.74% versus 54.86%. We attribute this token inefficiency to the highly asymmetric structure of the dataset (250 tuples by 10 tuples). Because Table B contains only 10 concepts, ABJ covers the entire 2,500-pair search space in as few as 13 LLM calls, making it exceptionally token-efficient. Under these conditions, the overhead of the Cluster Join pipeline—embedding generation, k-means partitioning, and the sample-based cluster filter step—consumes more tokens than the brute-force alternative. Notably, the tuned configuration used a filtering threshold of 0.0, meaning no cluster pairs were pruned by the sample-based filter. This is consistent with the dataset's design intent: since semantic similarity is a poor indicator of join relevance for Stack Overflow questions and programming concepts, the cluster filter's scoring mechanism provides no reliable signal for pruning, further compounding the pipeline overhead. This suggests that the clustering approach is better suited to larger, more symmetric datasets where pruning savings outweigh pipeline overhead, and whether a sufficiently large asymmetric table would shift this balance remains an open question.

*Classifier.* The Classifier strategy is most effective when a join reduces to a discrete label set. Thus, on the IMDb dataset where movie reviews carry either positive or negative sentiments, the

Classifier outperformed all other strategies in both F1-score and token expenditure, achieving an F1 of 78.08% while only using 22,362 tokens. Compared to the ABJ on IMDb (70,446 tokens, F1-score 49.92%), it reduces token expenditure by 68.3% while simultaneously improving F1-score by 28.16 percentage points. Furthermore, when compared to the next most token-efficient method for this dataset, Cluster Join (38,624 tokens), the Classifier Join not only reduces token cost by an additional 42% but also yields a substantial 22.52 percentage point increase in F1-score. On Stack Overflow, using the 10 programming concepts as the labels, the Classifier achieved an F1 of 74.95% with 30,216 tokens, the highest F1 among the strategies again. This demonstrates that when a join reduces to a discrete label set, a dedicated classifier approach significantly optimizes both accuracy and cost.

*Featurized-Decomposition Join.* FDJ's efficiency comes from a CNF filter over feature distance which prunes most pairs before LLM verification. When this filter extracts strong features, FDJ is extremely efficient. However, if the filter cannot form a useful discriminator, FDJ has very few pairs to prune and must verify every remaining candidate pair directly with the LLM, reverting to $O(M \times N)$ cost. We observed this across all of our datasets, the extracted feature distances were not a reliable signal for the join predicate, so the signal pruned little and fell back to verifying almost all pairs with the LLM.

*Advisor Routing.* The advisor routes each join to either the Classifier or Cluster Join strategy based on the predicate and a sample of each table, and identified all three datasets correctly on its first pass (Table 3). It recognized the sentiment join on IMDb and the tag taxonomy on Stack Overflow as a same-label classification, and the Emails contradiction, which requires reading both rows together, as a Cluster Join. These choices match the empirically optimal strategies in Table 2. The advisor's overhead is also negligible at 13,084 tokens across all three datasets combined relative to the tens of thousands of tokens for even the cheapest join configuration.

**Table 3: LLM Advisor Routing Decisions**

| Dataset | Strategy | Labels | Projection | Advisor Tokens |
|---|---|---|---|---|
| IMDb | Classifier | Positive, Negative | — | 9,418 |
| Emails | Cluster Join | — | No | 2,230 |
| Stack | Classifier | 10 concept tags | — | 1,436 |
| | | | **Total** | **13,084** |

## 4.4 Parameter Studies

*4.4.1 Cluster Join Configuration.* To comprehensively evaluate the performance of our cluster join methodology, we benchmarked our cluster join pipeline across all three datasets (Emails, Stack Overflow, and IMDb) using a rigorous grid search. We tested four distinct cluster ratios (0.025, 0.05, 0.075, and 0.1), which govern the cluster density within the tables. Specifically, the number of clusters generated for a given table was determined by scaling the dataset cardinality by the cluster ratio, bounded by a minimum of 2 clusters (calculated as $\max\{2, cardinality \times ratio\}$). For each generated

cluster configuration, we further evaluated the join performance across 100 distinct similarity thresholds, sweeping from 0.0 to 1.0 in increments of 0.01.

Given this large search space, establishing the single "Tuned Cluster Join" configuration reported in Table 2 required a systematic trade-off analysis between accuracy and computational cost. Our selection methodology proceeded as follows: first, we executed a complete search to identify the baseline configuration that yielded the absolute highest F1-score. Next, we isolated a candidate pool of alternative configurations that achieved an F1-score within 2.5 percentage points of this maximum.

From this subset, we conducted a subjective trade-off analysis. If an alternative configuration maintained an F1-score highly competitive with the maximum but offered substantial token cost savings, operationally defined as approximately a 10% or greater reduction in token expenditure, it was selected as the tuned configuration. Conversely, if the candidate configurations only offered negligible token savings (e.g., 1% reduction) at the cost of accuracy, we rejected the alternatives and defaulted to the configuration with the highest absolute F1-score. This approach ensures that the tuned cluster joins presented in our results represent the most practical balance of high semantic matching accuracy and LLM operational efficiency.

### 4.4.2 *Block Size Sensitivity.*

Our Cluster Join's matching stage batches tuples within each cluster pair and therefore requires a batch size. To choose a good value, we ran a simple fixed-block-size block join—a stripped-down block nested-loop evaluation with a single, manually set batch size—over the full tables and swept the block size, measuring the resulting accuracy and token cost. This is a tuning procedure, not part of our evaluation: the simple block join is used only to select a batch size and is not a baseline. Our baseline is ABJ, which sizes its batches automatically. Because our largest table is 250 rows, we tested block sizes of 5, 10, 15, 20, and 25 (Table 4).

Across all three datasets, block size 10 achieved the highest F1-score, while block size 5 maximized recall at substantially higher token cost. Beyond block size 15, F1-scores declined more steeply without proportional token savings. We use block size 15 for the Cluster Join matching stage; relative to block size 10 it reduces token expenditure by 21–33% on Emails and IMDb and changes F1 by 3–5 percentage points.

### 4.4.3 *Projection.*

The advisor enables projection on a per-join basis. To characterize its effects independently, we forced it on across all three datasets (Table 5). Projection increased Cluster Join F1 on all three datasets: by 4.52 points on IMDb (55.56% to 60.08%), 2.56 points on Emails (75.11% to 77.67%), and 0.41 points on Stack Overflow (69.74% to 70.15%). It also increased token expenditure, by 51.1% on IMDb (38,624 to 58,360 tokens), 22.6% on Emails (26,963 to 33,071 tokens), and 67.2% on Stack Overflow (76,942 to 128,617 tokens). Projection thus yields a consistent F1 gain at a consistent token cost for our tests; whether that exchange is worthwhile depends on the value a given deployment places on accuracy relative to token expenditure.

**Table 4: Block Size Exploration (averaged across 3 trials)**

| Dataset | Block | Recall (%) | Precision (%) | F1 (%) | Tokens |
|---------|-------|-----------|---------------|--------|--------|
| **IMDb** | 10x10 | 54.56 | 71.66 | 61.95 | 56,888 |
| | 15x15 | 45.11 | 78.90 | 57.32 | 44,951 |
| | 20x20 | 32.40 | 79.16 | 45.94 | 33,310 |
| **Emails** | 10x10 | 90.18 | 75.38 | 82.11 | 63,186 |
| | 15x15 | 83.94 | 74.47 | 78.92 | 42,219 |
| | 20x20 | 83.19 | 78.40 | 80.72 | 29,513 |
| **Stack** | 10x10 | 61.21 | 77.46 | 68.38 | 33,891 |
| | 15x10 | 55.15 | 79.00 | 64.95 | 31,866 |
| | 20x10 | 55.52 | 78.02 | 64.87 | 30,941 |

**Table 5: Effect of projection on Cluster Join (averaged over three trials)**

| Dataset | Configuration | Recall (%) | Precision (%) | F1 (%) | Tokens |
|---------|--------------|-----------|---------------|--------|--------|
| **IMDb** | Cluster Join | 41.39 | 84.45 | 55.56 | 38,624 |
| | Cluster Join + Projection | 47.62 | 81.37 | **60.08** | 58,360 |
| **Emails** | Cluster Join | 73.48 | 76.81 | 75.11 | 26,963 |
| | Cluster Join + Projection | 77.47 | 77.87 | **77.67** | 33,071 |
| **Stack** | Cluster Join | 63.27 | 77.68 | 69.74 | 76,942 |
| | Cluster Join + Projection | 68.36 | 72.03 | **70.15** | 128,617 |

## 5 CONCLUSION

Traditional relational databases struggle with unstructured data, and naive LLM-based semantic joins are computationally prohibitive. We presented SemJoin, an LLM-agent pipeline that treats strategy selection itself as the problem: rather than committing to a single fixed algorithm, an advisor routes each join to the execution strategy best suited to the data, falling back deterministically whenever its decision is uncertain. The central empirical finding is that this routing pays off. Across three datasets the routed strategy exceeded ABJ's F1-score on every workload while reducing token cost on two of the three, and it exceeded FDJ's F1 across all three at far lower token cost. Crucially, no single strategy won everywhere: the Classifier was best when a join reduces to a shared discrete label set (on IMDb cutting token cost by 68% and raising F1 by 28 percentage points over ABJ), while the Cluster Join was the better general-purpose choice for predicates that do not, and the advisor selected the empirically optimal strategy on all three workloads at negligible overhead—evidence that the routing decision can itself be delegated to the LLM cheaply. The optional projection step traded a consistent token-cost increase for a small, consistent F1 gain on all three datasets.

Having validated this approach on workloads of modest scale, including two custom-adapted datasets, the next step is to investigate whether these efficiencies hold under the pressures of larger, highly varied production workloads. Future research will build upon this foundation by scaling the evaluation, refining the advisor mechanism, and integrating additional latest LLMs.

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
