# OpenReview forum: "SemJoin: Semantic Join Optimization"
_VLDB.org/2026/Workshop/NOVAS — NOVAS 2026_

### Official Review · Reviewer_SaVU · 2026-07-07

**Confidence:** 4

**Improvement Opportunities:**

1. The main concern I have is that the tables used in experiments are a bit small. Similarly, we only know that the agent does a good job in 3 (albeit diverse) datasets. It would be useful to have some more evidence of robustness. This need not be new datasets. You can subsample on existing datasets, try other borderline/adversarial predicates etc

2. The other issue is that the current experiments operate at the extremes where one of the strategies is clearly optimal. It would be useful to test some cases where there is some ambiguity.

3. The reported cost model is a bit simplistic. It would be useful to report other metrics such as latency and cost of other operations (clustering, embedding, etc) etc

4. It is not clear which LLMs were used in the pipeline. Please mention it.

5. It is also not clear to me how the ground truth were constructed.

**Minor Comments:**

NA

**Short Summary:**

The paper tackles an interesting variant of SemJoin problem. Instead of sticking to a single strategy, the authors propose a agent based advisor that dynamically routes to one of two strategies based on join predicate and table samples with a deterministic fallback. The paper evaluates on few datasets and shows the agent can select the empirically optimal strategy resulting in lower cost.

**Strong Points:**

1. The problem variant is well motivated. It is obvious in hindsight that there will be no single fixed strategy is optimal across heterogeneous predicate/data types or samples. The experimental results shows some promising preliminary results. I also liked that every advisor decision has a deterministic fallback. The two strategies are also interpretable.

2. The experimental design is good and has promising results. The Stack Overflow dataset is promising and could stress test SemJoin works. The authors also conduct useful ablation studies.

3. I also liked that the paper briefly discusses some negative results. Overall, the paper is well situated in a crowded area.

---

### Official Review · Reviewer_FEQ9 · 2026-07-08

**Confidence:** 4

**Improvement Opportunities:**

O1. The method is somewhat heuristic. There are many moving parts in the proposed pipeline: the LLM advisor, embedding model, projection decision, and clustering configurations. However, the paper does not provide enough analysis of how robust these decisions are to prompt variation, sample selection, dataset size, or different LLMs. In particular, it is unclear whether the advisor is genuinely learning useful physical optimization signals. While Section 4.4 provided some parameter studies, more ablation studies are needed.

**Minor Comments:**

Perhaps the author should also consider the AI JOIN queries in Sembench or include other datasets to cover a wider range of join selectivities.

**Short Summary:**

In this paper, the authors propose SemJoin, an LLM-agent-based pipeline for optimizing semantic joins over unstructured data. Instead of using a single fixed join strategy, the system uses an LLM advisor to route each semantic join to either a Classifier strategy, predicate reduced to a shared discrete label set, or a Cluster Join strategy, based on embeddings, clusters and batch prompting. The paper evaluates SemJoin on IMDb reviews, email contradiction detection, and Stack Overflow question-tag matching, and compares against ABJ and FDJ baselines.

**Strong Points:**

S1. The paper studies an important and practical problem. AI joins are increasingly relevant in AI-native data systems, but naive pairwise LLM evaluation is too expensive!

S2. The core idea of routing between different physical strategies is intuitive. The distinction between classification-style semantic joins and more general pairwise semantic joins is reasonable, and the LLM advisor provides a simple mechanism for choosing between them. This direction is also related to recent work on physical implementations of semantic operators, such as semantic ORDER BY.

S3. The evaluation includes meaningful comparisons with recent semantic join baselines, including ABJ and FDJ. The reported results suggest that the routed strategy can improve F1 over ABJ across datasets and achieve much lower token cost than FDJ in the evaluated settings.

---

### Official Review · Reviewer_s1XE · 2026-07-09

**Confidence:** 4

**Improvement Opportunities:**

O1: The main problem definition is not sufficiently clear. Throughout the paper, it is difficult to understand whether the goal is to optimize the execution of semantic joins or to identify semantically matching values across datasets. The title and framing suggest a join optimization problem, while many of the examples and methods resemble entity matching or semantic matching tasks. The paper would greatly benefit from a precise definition of semantic join, its objectives, and how it differs from related tasks such as entity matching and record linkage.

O2: The connection to entity matching literature is missing. Many of the examples, motivations, and matching decisions appear very similar to entity matching. If the key novelty is that joins are defined through natural-language predicates rather than schema-level correspondence, this distinction should be clearly articulated. Entity matching should at least be discussed in the related work section and contrasted with the proposed setting.

O3: The motivation around integrating unstructured data is not clear and seems irrelevant. The paper repeatedly emphasizes challenges associated with integrating unstructured data, yet the evaluated tasks are formulated as joins between two tables. It remains unclear how the proposed method specifically addresses challenges unique to unstructured data integration rather than semantic matching between structured records.

O4: Figure 1 requires substantial improvement. As the main methodological figure, it appears unpolished and is difficult to follow.

O5: The running examples can be strengthen. The shirt-and-pants example is not particularly convincing and does not help clarify the problem setting. Given the datasets used in the paper, the authors could use more realistic and informative examples. Furthermore, the current example raises questions such as: what if the user wants to join on brand?

O6: The evaluation section needs additional discussion regarding the comparison to ABJ. Some reported results appear inconsistent with those reported in the ABJ paper. Since ABJ is a primary baseline, the paper should carefully explain implementation details and why the results differ from the original paper.

**Minor Comments:**

See above.

**Short Summary:**

This paper presents SemJoin, an LLM-based framework for semantic joins that covers a classifier-based strategy and a clustering-based strategy depending on the join predicate and input data. An LLM advisor determines the most suitable execution plan, aiming to reduce the cost of semantic joins while maintaining accuracy. Experiments on three datasets show improvements over recent semantic join baselines.

**Strong Points:**

S1: The problem of optimizing semantic joins with LLMs is timely and relevent for the workshop.
S2: Selecting different execution strategies depending on workload characteristics is intuitive and potentially useful.
S3: The evaluation considers multiple (3) datasets and compares against relevant recent baselines (ABJ and FDJ).

---

### Decision · Program_Chairs · 2026-07-16

**Decision:**

Accept

**Comment:**

SemJoin addresses the timely problem of reducing the cost of semantic joins by selecting between classifier-based and clustering-based execution strategies. Its adaptive routing approach is intuitive, interpretable, and supported by promising results across multiple datasets and recent baselines. We believe this work will generate valuable discussion on physical optimization strategies for semantic operators at the workshop.